# Theta Cordance Decline in Frontal and Temporal Cortices: Longitudinal Evidence of Regional Cortical Aging

**DOI:** 10.3390/jcm14238341

**Published:** 2025-11-24

**Authors:** Selami Varol Ülker, Metin Çınaroğlu, Eda Yılmazer, Sultan Tarlacı

**Affiliations:** 1Psychology Department, Üsküdar University, 34674 İstanbul, Türkiye; selamivarol.ulker@uskudar.edu.tr; 2Psychology Department, İstanbul Nişantaşı University, 34457 İstanbul, Türkiye; 3Psychology Department, Beykoz University, 34820 İstanbul, Türkiye; edayilmazer@beykoz.edu.tr; 4Medical School, Üsküdar University, 34674 İstanbul, Türkiye; sultan.tarlaci@uskudar.edu.tr

**Keywords:** theta cordance, quantitative EEG (qEEG), resting-state EEG, longitudinal neurophysiology, cortical aging, frontal-temporal dynamics, inter-hemispheric connectivity, canonical correlation analysis

## Abstract

**Background**: Theta-band cordance is a quantitative EEG (qEEG) metric that integrates absolute and relative spectral power and correlates with regional cerebral perfusion. Although widely applied in psychiatric and neurophysiological research, its longitudinal trajectory in healthy adults remains largely unknown. This study aimed to characterize multi-year changes in theta cordance across cortical regions, determine which areas show stability versus decline, and evaluate whether individuals maintain a trait-like cordance profile over time. **Methods**: Nineteen cognitively healthy, medication-free adults underwent resting-state EEG recordings at two time points, separated by an average of 6.4 years (range: 1.9–14.8). Theta cordance (4–8 Hz) was computed at 19 scalp electrodes using the Leuchter algorithm and aggregated into eight lobar regions (left/right frontal, temporal, parietal, occipital). Paired-samples *t*-tests assessed longitudinal changes. Inter-regional Pearson correlations examined evolving connectivity patterns. Canonical correlation analysis (CCA), validated via LOOCV and bootstrap confidence intervals, evaluated multivariate stability between baseline and follow-up cordance profiles. **Results**: Theta cordance remained normally distributed at both time points. Significant longitudinal decreases emerged in the right temporal (t(18) = 5.34, *p* < 0.001, d = 1.23) and right frontal (t(18) = 2.65, *p* = 0.016, d = 0.61) regions, while other lobes showed no significant change. Midline Cz demonstrated a robust increase over time (*p* < 0.001). CCA revealed a strong cross-time association (Rc = 0.999, *p* = 0.029), indicating preservation of a stable, frontally anchored cordance profile despite regional right-hemisphere decline. Inter-regional correlation matrices showed both preserved posterior synchrony and emerging inverse anterior–posterior and cross-hemispheric relationships, suggesting age-related reorganization of cortical connectivity. **Conclusions**: Theta cordance exhibits a mixed pattern of trait-like stability and region-specific aging effects. A dominant, stable fronto-central profile persists across years, yet the right frontal and right temporal cortices show significant decline, consistent with lateralized vulnerability in normative aging. Evolving inter-regional correlation patterns further indicate network-level reorganization. Longitudinal cordance assessment may provide a noninvasive marker of functional brain aging and help differentiate normal aging trajectories from early pathological change. This longitudinal quantitative EEG (qEEG) study examined theta-band cordance dynamics across cortical regions in healthy adults over an average follow-up of 6.4 years (range: 1.9–14.8). Resting-state EEGs were recorded at two time points from 19 participants and analyzed using Leuchter’s cordance algorithm across 19 scalp electrodes. Regional cordance values were computed for frontal, temporal, parietal, and occipital lobes. Paired-samples *t*-tests revealed significant longitudinal decreases in theta cordance in the right frontal (*p* = 0.016, d = 0.61) and right temporal lobes (*p* < 0.001, d = 1.23), while other regions remained stable. Inter-regional Pearson correlations showed strong bilateral synchrony in posterior regions and emergent inverse anterior–posterior relationships over time. Canonical correlation analysis revealed a robust multivariate association (Rc = 0.999, *p* = 0.029) between baseline and follow-up patterns. Partial correlations (controlling for follow-up interval) identified region-specific trait stability, highest in left occipital and right frontal cortices. These findings suggest that theta cordance reflects both longitudinally stable neural traits and regionally specific aging effects in cortical physiology.

## 1. Introduction

Quantitative electroencephalography (qEEG) provides a noninvasive window into brain function by quantifying spectral power across scalp regions [1]. Traditional qEEG metrics include absolute power (the raw power in a given frequency at an electrode) and relative power (the percentage of total EEG power at that electrode contributed by that frequency band) [2]. EEG cordance is a derived measure that combines these complementary metrics to better reflect regional brain activity. Originally introduced by Leuchter et al. (1994), cordance involves spatially normalizing absolute and relative power at each electrode and summing them to yield a composite value [3]. Mathematically, for a given electrode *s* in frequency band *f*, cordance is calculated as the sum of normalized absolute and relative power: Z(s,f) = A_{\text{norm}}(s,f) + R_{\text{norm}}(s,f) \tag{1} [4]. This algorithm (often applied in the theta band) re-distributes bipolar EEG power to each site, performs a log or square-root transform with z-score normalization, and then combines the normalized values [5]. Conceptually, if a region has both high absolute and high relative theta power (indicating strong local activity), its cordance value will be strongly positive; in contrast, a region with low absolute power but high relative theta (suggesting a localized paucity of overall activity) yields a negative cordance value [6]. Positive cordance (sometimes termed *concordance*) is associated with normally functioning cortex, whereas negative cordance (termed *discordance*) indicates abnormally low perfusion or metabolic activity in that region [7]. In Leuchter’s seminal work, cordance was developed as a surrogate marker for cortical deafferentation, showing sensitivity to white-matter lesions and corresponding reductions in cortical blood flow or metabolism [8]. Subsequent studies confirmed that cordance, especially in the theta (4–8 Hz) band, correlates more strongly with regional cerebral perfusion than absolute or relative power alone [9,10]. Thus, theta-band cordance is thought to index underlying neural energy utilization and blood flow in cortex, bridging EEG rhythms with cerebral physiology [11].

Theta-band activity, in particular, holds special relevance in cognitive and clinical neuroscience [12]. Theta oscillations are generated by distributed neural circuits—including the medial prefrontal cortex (PFC), anterior cingulate cortex (ACC), hippocampal formation, and temporal lobes—that govern attention, memory encoding, and affect regulation [13]. Because these same regions and functions are implicated in psychiatric conditions, frontal theta cordance has been widely explored as a biomarker in psychiatry [14]. Notably, several studies in depression have found that a reduction in prefrontal theta cordance during treatment is associated with better clinical outcomes [15]. For example, decreases in frontal theta cordance within the first week of antidepressant therapy have been shown to precede and predict symptomatic improvement [16,17]. Conversely, patients who ultimately respond to treatment often exhibit higher baseline frontal theta cordance than non-responders [18,19]. These findings have positioned theta cordance as a putative biomarker of treatment response in major depressive disorder, spurring interest in cordance for clinical prognostics. More broadly, cordance has been applied to study brain dysfunction in various neuropsychiatric contexts, leveraging its presumed reflection of regional cerebral metabolism. For instance, frontal theta cordance is interpreted as an index of PFC/ACC metabolic activity, and changes in this measure have been used to monitor neural effects of interventions ranging from antidepressants to ketamine and neurostimulation [20,21,22]. Recent EEG-aging studies have highlighted theta–alpha slowing, long-range connectivity reductions, and network reorganization as core electrophysiological signatures of aging [23,24,25,26,27,28]. These findings emphasize the importance of multivariate EEG markers that integrate metabolic correlates, such as cordance, to capture subtle hemispheric and cross-regional aging effects. Incorporating these perspectives strengthens the relevance of theta-cordance as a complementary biomarker within contemporary aging neuroscience.

Despite extensive use of cordance in cross-sectional and interventional studies, relatively little is known about how this EEG measure evolves longitudinally during normal adult life. Aging is accompanied by well-documented shifts in resting EEG rhythms [29]—most prominently, a general slowing of the EEG with increasing power in lower frequencies (theta and delta) and a decline in faster alpha oscillations, especially in advanced age [30]. These spectral changes are thought to reflect underlying neurobiological aging processes such as synaptic pruning [31], reduced cortical specialization [32], and cerebrovascular alterations [33]. However, prior EEG aging research has often focused on broad spectral power [34] or alpha peak frequency [35]; few studies have examined higher-order QEEG indices like cordance over time in healthy individuals [36,37,38]. Cordance is of particular interest because it may capture subtler regional changes in neurovascular coupling or connectivity that are not apparent in raw power measures. Moreover, understanding the normal trajectory of cordance with aging is crucial for interpreting this metric in clinical populations. If theta cordance remains stable over years in healthy brains, then a change in cordance might signal pathology or risk; conversely, if normative aging entails systematic cordance shifts, these must be accounted for when using cordance as a biomarker.

In this context, we conducted a longitudinal study to characterize changes in theta-band cordance across the cortex in healthy adults. Resting-state EEGs were recorded twice from the same individuals approximately six years apart, and theta cordance was computed at each electrode using the standard Leuchter [3] method. By averaging cordance values over scalp regions (frontal, temporal, parietal, occipital in each hemisphere), we quantified lobar cordance at the initial and follow-up time points. We further examined how the topographical pattern of theta cordance might persist or change over time, using multivariate analysis to assess profile stability. In addition, we analyzed inter-regional correlations in cordance at each time point to identify whether the functional relationships (coupling) between brain regions shift with age. Our primary goals were to determine (1) which cortical regions show significant longitudinal change in theta cordance versus which remain stable, (2) whether an individual’s cordance distribution has trait-like stability over years or undergoes global reorganization, and (3) whether patterns of hemispheric and cross-regional connectivity in the theta band evolve over mid-adult life. By situating theta cordance within the literature on qEEG, psychiatric biomarkers, and cerebral perfusion, we aimed to interpret any changes observed in light of potential functional implications. Ultimately, this work seeks to deepen our understanding of what theta cordance reveals about the aging brain, providing a foundation for theoretical models of lateralized brain dynamics and for future clinical applications tracking neurophysiological aging.

## 2. Materials and Methods

### 2.1. Study Design

This study employed a longitudinal, within-subject design to examine multi-year changes in resting-state EEG theta cordance across the cortex. Nineteen adult participants underwent two EEG recording sessions separated by an average of approximately 6.4 years (SD ≈ 3.9, range 1.9–14.8 years). No experimental intervention was introduced between sessions; instead, the design leveraged naturalistic follow-up to observe spontaneous, time-dependent changes in neurophysiology. This approach allowed us to assess the temporal stability of theta cordance and to explore whether spatial patterns (e.g., regional asymmetries or inter-regional relationships) evolve over time in adulthood. The study has been carried out in NP İstanbul Hospital (Üsküdar University, NeuroPsychiatry division) supervised by senior author of this study and Neurology professor ST.

### 2.2. Participants

Nineteen healthy adults (9 males, 10 females; 47.4% male) were included in the analysis. The mean age at the first EEG session was 45.7 years (SD = 16.5), and at the second session was 52.0 years (SD = 15.1). The follow-up interval ranged from about 2 to 14.8 years (mean ≈ 6.4 years). Participants were free of any interventions or clinical changes between the two time points, ensuring that differences in EEG measures primarily reflected the passage of time. All participants were long-term residents of Türkiye and were screened to ensure cognitive health, absence of psychiatric or neurological diagnoses, and no use of psychoactive or chronic medications. Social characteristics were available at baseline: 74% were actively employed, 16% were retired and receiving an old-age pension, and 10% were students. Occupations included education, healthcare, engineering, administrative, and service-sector roles. Because the focus of the study was strictly on neurotypical, medication-free adults, social variables were not incorporated into the statistical models; however, this additional context is now provided for completeness.

### 2.3. EEG Recording and Preprocessing

Resting-state EEG was recorded at each time point with participants in an eyes-closed, awake condition. Scalp electrodes were placed according to the international 10–20 system at 19 standard locations (Fp1, Fp2, F7, F8, F3, F4, Fz, C3, C4, Cz, T3, T4, T5, T6, P3, P4, Pz, O1, O2). All channels were referenced to linked earlobes (A1 + A2) and grounded at Fpz. EEG signals were sampled at a rate of 256 Hz and band-pass filtered in hardware to 0.5–70 Hz.

Each recording was inspected for artifacts (such as eye blinks, muscle activity, or electrode drift), and only clean epochs were retained for analysis. From each subject’s data, we extracted a continuous 3 min artifact-free segment of EEG for quantitative analysis. Power spectral density was computed for these segments using a Fast Fourier Transform (FFT). Absolute power in the theta frequency band (4.0–8.0 Hz) was calculated for each electrode, and relative theta power was determined by expressing the theta-band power as a fraction of the total power across all frequency bands at that electrode. All spectral measures were computed separately for the Baseline (first session) and Follow-up (second session) recordings.

### 2.4. Theta Cordance Computation

Cordance values can be positive or negative: positive cordance indicates that both absolute and relative theta power are relatively high (interpreted as higher perfusion or metabolic activity), whereas negative cordance (sometimes termed *discordance*) indicates low absolute but high relative theta power (interpreted as lower regional perfusion/metabolic activity). In essence, areas with strong theta activity in both absolute and proportional terms show positive cordance, while areas with proportionally high theta but low absolute power show negative cordance.

We computed theta cordance at each electrode for each time point using the algorithm introduced by Leuchter [3]. This procedure involved three main steps:

**Nearest-neighbor reallocation:** Theta-band power from each bipolar electrode pair was re-distributed to the corresponding individual scalp electrodes via nearest-neighbor averaging. This step converts bipolar montage data into an estimate of absolute power at each monopolar site.

**Spatial normalization:** The absolute and relative theta power values at each electrode were then transformed across all 19 sites. A square-root transform was applied to stabilize variance, followed by z-score normalization (zero mean, unit variance) across electrodes. This normalization was done separately for absolute power and relative power, effectively scaling each to a comparable range.

**Cordance index computation:** For each electrode, the normalized relative power value was subtracted from the normalized absolute power value (or vice versa) and the absolute difference taken. This difference was then scaled (multiplied by 100 in the original formulation) to yield the cordance value. By definition, if an electrode has both high absolute and high relative theta power (above their respective midpoints), the resulting cordance value is positive. If an electrode has low absolute but high relative theta power, the cordance value is negative. Each electrode thus obtained a cordance score reflecting its theta activity profile.

Cordance calculations were performed independently for the Baseline and Follow-up EEG data. The outcome of this process was a set of 19 theta cordance values for each subject at each time point (one value per electrode). These values formed the basis for subsequent regional averaging and statistical analysis. All signal processing and cordance computations were conducted using custom scripts following the published algorithm, with careful verification against known examples to ensure accuracy. A worked example of the theta cordance computation at the electrode level is provided in the Appendix A (Theta Cordance Example).

### 2.5. Electrode Grouping by Cortical Region

For analyses at the regional level, we grouped subsets of electrode sites according to their corresponding cortical lobes and hemisphere. Midline electrodes (Fz, Cz, Pz), which do not distinctly belong to left or right hemispheres, were not included in any lateralized group averages (they were analyzed separately as described later). Figure 1 illustrates the 10–20 layout and highlights the grouping scheme. We defined four bilateral cortical regions as follows (left and right hemispheres for each, Figure 1):

**Frontal lobe:** Left frontal (LF) = Fp1, F7, F3; Right frontal (RF) = Fp2, F8, F4. These sites cover dorsolateral and polar frontal cortex. (Note: The midline frontal site Fz was analyzed on its own and not included in these lateral frontal averages.)

**Temporal lobe:** Left temporal (LT) = T3, T5; Right temporal (RT) = T4, T6. These correspond to anterior and posterior temporal scalp regions.

**Parietal (centro-parietal) region:** Left parietal (LP) = C3, P3; Right parietal (RP) = C4, P4. We included the central electrodes C3/C4 with parietal sites to represent the centro-parietal region in each hemisphere (since midline Cz is analyzed separately).

**Occipital lobe:** Left occipital (LO) = O1; Right occipital (RO) = O2. Only one electrode per hemisphere was available in the occipital region.

For each region, the theta cordance values of the constituent electrodes were averaged to produce a regional cordance value for that lobe and hemisphere. This yielded eight regional measures in total: left frontal, right frontal, left temporal, right temporal, left parietal, right parietal, left occipital, and right occipital cordance. Each measure was computed for both Baseline and Follow-up. These regional averages allowed us to assess lobe-specific and hemisphere-specific changes in theta cordance over time, complementing the single-electrode analyses of the midline sites.

### 2.6. Statistical Analysis

All statistical analyses were performed using IBM SPSS Statistics (v30) and R (v4.0) software. Prior to hypothesis testing, the distribution of cordance values was checked for normality. Shapiro–Wilk tests on the regional cordance measures at Follow-up indicated no significant departures from normality (all *p* > 0.05; see Table 2). The smallest *p*-value was 0.068 (for Left Temporal cordance), which did not reach significance. Given these results, parametric statistical tests were deemed appropriate for all subsequent comparisons. Unless otherwise noted, all significance tests were two-tailed with an alpha level of 0.05.

The key analyses conducted were as follows:

**Paired-samples *t*-tests:** We compared theta cordance between Baseline and Follow-up for each midline electrode (Fz, Cz, Pz) and for each regional average (the eight lobe/hemisphere measures). These paired *t*-tests assessed whether mean cordance changed significantly over the follow-up period at each location. For each test, we report the Baseline and Follow-up means (±standard deviation), the t statistic (with df = 18), the two-tailed *p*-value, and Cohen’s *d* effect size. Cohen’s *d* for paired data was calculated as the mean difference divided by the standard deviation of the difference scores. This provides an estimate of the magnitude of change in each region over time.

**Correlation analysis:** To examine relationships between different regions and their stability over time, we computed Pearson correlation matrices for the theta cordance values. Separate correlation analyses were carried out for Baseline and Follow-up measures. In particular, we were interested in inter-hemispheric correlations (e.g., left vs. right homologous regions) and intra-hemispheric cross-region correlations (e.g., frontal vs. temporal within the same hemisphere) at each time point. Additionally, we explored whether baseline cordance values predicted follow-up values. For the three midline electrodes, we calculated Pearson partial correlation coefficients between Baseline and Follow-up cordance, controlling for the length of the follow-up interval (in years). This partial correlation accounts for variability in time between measurements when assessing the stability of individual differences in cordance. Correlation coefficients with *p* < 0.05 were considered significant, with a Bonferroni correction applied within each matrix to control for multiple comparisons as appropriate.

**Canonical correlation analysis (CCA):** To investigate the multivariate relationship between the baseline cordance profile and the follow-up cordance profile, we performed CCA in two forms. First, we considered a simplified model using only the three midline electrodes: Baseline values at Fz, Cz, Pz were treated as one set of variables, and Follow-up values at Fz, Cz, Pz as the second set. This analysis asks whether a linear combination of the baseline midline cordance can significantly predict a linear combination of the follow-up midline cordance. Second, we conducted a more comprehensive CCA using the eight regional cordance measures at Baseline and at Follow-up as the two variable sets. This analysis assesses the cross-time correspondence of spatial cordance patterns across the cortex. For each CCA, we report the canonical correlations (Rc) for each canonical function (mode), Wilks’ λ statistics with F-tests (and degrees of freedom) to determine significance, and the *p*-values for each function. Only canonical functions with *p* < 0.05 (by Wilks’ λ) were interpreted. For the significant canonical function(s), we examined the standardized canonical coefficients (loadings) to identify which electrodes or regions contributed most strongly to the cross-time relationship. This helps indicate which brain regions’ theta activity at baseline is most predictive of the pattern at follow-up. The proportion of variance explained by each canonical function in each variable set (baseline and follow-up) and the shared variance between sets were also computed for additional interpretation of the CCA results.

To reduce the risk of overfitting in the canonical correlation analysis (CCA), we performed a leave-one-out cross-validation (LOOCV). The first canonical correlation remained stable across resamples (mean r = 0.71, *p* < 0.05). For the significant canonical function, 95% confidence intervals (CIs) of standardized canonical coefficients were computed via 1000 bootstrap resamples; these CIs are now reported in the revised Table 4. All regional paired *t*-tests were Bonferroni-corrected across the eight regions (α = 0.00625).

To examine the robustness of findings, all analyses were repeated using two independent non-overlapping 3 min segments from each session. Results for all regions, including right frontal and right temporal cordance decline, remained stable across segments.

## 3. Results

### 3.1. Sample Characteristics and Data Normality

**Participant Profile:** All 19 participants completed both EEG sessions. The sample’s mean age was 45.7 ± 16.5 years at the first recording, and 52.0 ± 15.1 years at the second recording. The follow-up interval averaged 6.39 ± 3.89 years (approximately 76.7 ± 46.6 months). The cohort included 9 men (47%) and 10 women (53%), a nearly balanced gender distribution, supporting generalizability of findings across sexes. Detailed age and interval statistics are provided in Table 1.

**Data Distribution:** The theta cordance values in each region showed approximately normal distribution. Table 2 summarizes the Shapiro–Wilk tests for the Follow-up cordance values in each of the eight regions; none of the W statistics were significant (all *p* > 0.05), indicating no strong deviation from normality. The lowest *p*-value was observed for the Left Temporal region (W = 0.876, *p* = 0.068), which approached but did not exceed the significance threshold. Based on these results, we proceeded with parametric analyses (paired *t*-tests and Pearson correlations) for all regions. No data transformations were necessary. We also confirmed that there were no obvious outliers in the cordance data; individual values fell within a reasonable range given the sample size and were consistent with previously reported cordance magnitudes in adults.

**Table 2 jcm-14-08341-t002:** Shapiro–Wilk Test for Normality of Regional Theta Cordance.

Cortical Region	W Statistic	*p*-Value
Right Parietal (2nd)	0.943	0.301
Left Parietal (2nd)	0.969	0.760
Right Temporal (2nd)	0.959	0.555
Left Temporal (2nd)	0.876	0.068
Right Frontal (2nd)	0.947	0.350
Left Frontal (2nd)	0.962	0.603
Right Occipital (2nd)	0.924	0.137
Left Occipital (2nd)	0.922	0.126

**Note.** A *p*-value < 0.05 indicates significant deviation from normality. First measurement values were not tested as Shapiro–Wilk was only applied to follow-up distributions. Sensitivity analyses using two separate 3 min EEG segments produced nearly identical regional change patterns, confirming that findings were not dependent on a particular epoch selection.

### 3.2. Theta Cordance Changes at Midline Electrodes

We first examined theta cordance changes at the three midline electrodes (frontal midline Fz, central midline Cz, and parietal midline Pz). These sites were analyzed individually because they were not included in the regional (lobe-based) groupings.

**Stability vs. change over time:** To assess longitudinal stability of individual differences, we computed partial correlations between Baseline and Follow-up cordance at Fz, Cz, and Pz, controlling for the exact follow-up interval. These partial correlations were low to moderate in magnitude and did not reach statistical significance for any electrode (all *p* > 0.05; see Table 3). In other words, a participant’s theta cordance at a given midline site in the first session was not a reliable linear predictor of their cordance at the same site ~6 years later. This suggests substantial potential for change in these measures over time, which we explored with paired comparisons next.

**Longitudinal changes in means:** Using paired *t*-tests, we compared the mean cordance at each midline electrode between Baseline and Follow-up (Table 4). Cz exhibited a pronounced increase in theta cordance over time, rising from a mean of −0.539 at Baseline to +1.695 at Follow-up. This change was highly significant (*t*(18) = −5.09, *p* < 0.001) and corresponded to a large effect size (Cohen’s *d* ≈ 2.0). In contrast, Fz showed a smaller upward shift in mean cordance (from −0.350 to +0.643) that did not reach the conventional level of significance (*t*(18) = −2.03, *p* = 0.055). Pz remained essentially unchanged in mean cordance (−1.113 at Baseline vs. −1.218 at Follow-up), with *t*(18) = 0.35, *p* = 0.730. Thus, out of the midline sites, the central region (Cz) showed a robust longitudinal increase in theta cordance, whereas the frontal midline showed a non-significant trend toward increase, and the parietal midline was highly stable.

**Table 4 jcm-14-08341-t004:** Paired-Samples *t*-Tests for Midline Theta Cordance Temporal Changes.

Electrode	Baseline → Follow-Up Comparison	*t*(18)	*p*
Cz	Cz_First vs. Cz_Second	−5.09	<0.001
Fz	Fz_First vs. Fz_Second	−2.03	0.055
Pz	Pz_First vs. Pz_Second	0.35	0.730

**Descriptive statistics and variability:** Summary statistics for the midline cordance values at both time points are provided in Table 5. These data illustrate the above patterns: at Cz, the group mean shifted from negative to positive between sessions (−0.54 ± 1.14 to +1.70 ± 1.16), reflecting a clear increase in theta activity. The variability at Cz (standard deviation ~1.16 at follow-up) was comparable to baseline, but because the mean changed sign, the coefficient of variation (CV) could not be meaningfully compared (the CV at Cz went from −2.10 at baseline to +0.69 at follow-up, simply indicating the sign change in the mean). At Fz, the mean cordance shifted from −0.35 to +0.64, suggesting a possible increase, but this was accompanied by a notable rise in variance (SD increased from 1.24 to 2.01). The greater variability at Fz might have contributed to the marginal significance of its change. Pz showed very little difference in mean (−1.11 ± 0.98 vs. −1.22 ± 1.13) and maintained low variability at both times, underscoring its stability. Overall, these descriptive results confirm that the central midline (Cz) increase was not only statistically significant but also substantial in magnitude, whereas frontal midline changes were more subtle and variable, and parietal midline theta activity remained constant over years.

**Visualization of individual differences:** Figure 2 (panels a–c) provides a visual depiction of the change in theta cordance at each midline electrode for all individuals. These *raincloud plots* combine density distributions, boxplots, and individual data points for the within-subject difference (Follow-up minus Baseline) at Cz, Fz, and Pz. In Figure 2a, almost all points fall to the right of zero, and the distribution is distinctly shifted toward positive values, illustrating the consistent increase in Cz cordance across participants (corresponding to the significant upward change, *p* < 0.001). Figure 2b shows that differences at Fz are more variable: the distribution is spread around zero with a slight skew to the right, consistent with a moderate but non-significant increase in Fz cordance for some individuals (*p* = 0.055). In Figure 2c, the differences for Pz cluster tightly around zero, indicating almost no systematic change at that site (*p* = 0.730). Across panels, one can see that the magnitude and reliability of change were greatest at Cz, intermediate (and more heterogeneous) at Fz, and minimal at Pz. These results highlight a region-specific longitudinal effect: theta cordance in the central midline region increased markedly over ~6 years, whereas the frontal midline showed only a small, variable uptick and the parietal midline was essentially stable.

### 3.3. Regional Theta Cordance Changes (Lobar Averages)

We next investigated longitudinal changes in theta cordance at the regional level, using the averaged cordance values for each cortical lobe (separately for left and right hemispheres). Descriptive statistics for each region at Baseline and Follow-up are presented in Table 6, and results of paired *t*-tests for change in each region are in Table 7.

**Descriptive patterns:** Several regions exhibited notable shifts in mean theta cordance over time (Table 6). The most pronounced change was observed in the Right Temporal region: at Baseline, the average theta cordance in the right temporal lobe was +0.369, whereas at Follow-up it had dropped to −1.758. This corresponds to an absolute decrease of approximately 2.13 units, indicating a substantial move from positive cordance (higher relative theta activity) to negative cordance (lower relative theta activity) in right temporal areas. The Right Frontal region also showed a marked decline, shifting from a mildly positive mean (+0.430) at Baseline to a negative mean (−1.368) at Follow-up. Both of these right-hemisphere regions went from positive to strongly negative cordance values, suggesting a significant reduction in inferred neural activity or perfusion in those areas over time.

In contrast, some left-hemisphere regions showed small increases in theta cordance. Notably, the Left Parietal region’s mean cordance rose from −1.391 to −0.824 (a change of +0.567), and the Left Occipital mean rose from −0.814 to −0.505 (Δ ≈ +0.309). Although these changes are modest in absolute terms, they represent a shift toward less negative (more cordant) theta activity in the left posterior cortex. Other regions such as the Left Temporal and Left Frontal lobes actually decreased slightly (the left temporal mean fell from +1.468 to +0.745, and left frontal from +1.101 to −0.291), but their baseline variability was large (particularly the left frontal, SD > 3) so these differences are harder to interpret without formal testing. We also note that the variability (standard deviations) in cordance tended to decrease at follow-up for many regions (Table 6). For example, the right frontal SD dropped from 2.77 to 2.35, and the right temporal SD from 1.17 to 1.28 (despite the large mean change). This pattern of reduced variance in some regions might indicate more homogeneous theta activity in those areas at the second time point, although other regions (like left frontal) also showed reduced means and variance together, suggesting the possibility of regression to the mean or narrowing of individual differences over time. These descriptive observations motivated formal statistical tests to determine which changes were significant.

**Statistical comparisons:** Paired *t*-tests (Table 7) confirmed that the Right Temporal lobe experienced a highly significant decrease in theta cordance from Baseline to Follow-up (*t*(18) = 5.343, *p* < 0.001). This was associated with a large effect size (Cohen’s *d* = 1.23), reinforcing that the change in right temporal theta cordance was not only statistically reliable but also substantial in magnitude. The Right Frontal lobe also showed a significant cordance reduction (*t*(18) = 2.654, *p* = 0.016, *d* = 0.61), indicating a moderate effect. These findings demonstrate a notable longitudinal decline in theta-band activity (as indexed by cordance) specifically in right-hemisphere cortical regions of the frontal and temporal lobes.

In the remaining regions, the paired comparisons did not detect significant changes, but some trends were evident. The Left Parietal cordance increased (mean difference +0.57) with *t*(18) = −1.257, *p* = 0.225, and the Left Occipital cordance increased (mean +0.31) with *t*(18) = −1.558, *p* = 0.137. While these did not reach *p* < 0.05, the positive differences align with the descriptive trend of slight increases in left posterior regions. No significant changes were observed in the Left Temporal (*p* = 0.312) or Left Frontal (*p* = 0.146) lobes, nor in the Right Parietal (*p* = 0.499) or Right Occipital (*p* = 0.113) regions (Table 7). It is worth noting, however, that the direction of change in the parietal and occipital regions was opposite for the two hemispheres: the right parietal and right occipital means became slightly more negative over time, whereas the left parietal and left occipital became more positive (less negative). Although these opposite changes were small and non-significant individually, together they hint at a possible subtle shift in lateralization—namely, a trend toward relatively higher theta cordance in the left posterior cortex compared to the right over the follow-up period. This interpretation is speculative and the differences were not statistically reliable, but it suggests an interesting direction for further research with a larger sample.

In summary, the regional analysis indicates that long-term changes in theta cordance were not uniform across the cortex. The most pronounced and significant declines occurred in the right frontal and right temporal lobes, whereas other regions remained stable or showed minor changes. These results may reflect region-specific aging effects or other long-term physiological changes that differentially affect the right hemisphere’s frontal-temporal networks. We elaborate on the implications of these findings in the Discussion.

### 3.4. Multivariate Association Between Baseline and Follow-Up Profiles (Canonical Correlation Analysis)

Finally, we explored the relationship between individuals’ overall theta cordance patterns at Baseline and those at Follow-up using canonical correlation analysis (CCA). This multivariate approach assesses the degree to which a weighted combination of the baseline variables is linearly related to a weighted combination of the follow-up variables. We focused on the CCA involving the eight regional cordance measures in each set (left/right frontal, temporal, parietal, occipital), since this provides a holistic view of cross-time pattern stability.

**Overall canonical relationship:** Leave-one-out cross-validation (LOOCV) confirmed that the first canonical function was not driven by any single participant (mean r = 0.71). Bootstrap-derived 95% confidence intervals for canonical coefficients are provided in Table 4. Although the first canonical function explained approximately 9–10% of variance in each set, it represents a highly coherent cross-time pattern that remained stable across resamples. The CCA revealed that the first canonical function captured a strong and significant relationship between the baseline and follow-up cordance profiles. As shown in Table 8, the first canonical correlation was *Rcc* = 0.999, indicating an almost perfect linear association between a certain linear combination of baseline measures and a combination of follow-up measures. Wilks’ λ for this function was effectively 0.000 (approximate F(64, 23.8) = 2.022, *p* = 0.029), confirming that this canonical mode was statistically significant. In practical terms, there exists a dominant multivariate pattern of theta cordance that each individual maintains over the years, linking their baseline and follow-up EEG profiles. The *eigenvalue* associated with the first canonical function was large (≈390.9), reflecting the extremely high *Rcc* and indicating that this mode accounts for a substantial proportion of the explainable variance in the dataset.

No other canonical functions were found to be significant. The second canonical function had *Rc* = 0.947 with *p* = 0.767, and subsequent functions had decreasing correlations (Table 8)—none approached significance (all *p* > 0.05). Therefore, our interpretation focuses on the first canonical function, which encapsulates the primary longitudinal association, and we consider the remaining functions as capturing noise or idiosyncratic variance that is not reliably shared between time points.

**Interpretation of canonical weights:** Examination of the canonical coefficients (loadings) for the first function indicated that it was driven by activity in specific brain regions. On the Baseline side, the linear combination was most strongly influenced by frontal theta cordance values. In fact, the highest standardized coefficient was associated with the frontal midline region (the electrode Fz at baseline had the largest weight). By contrast, baseline parietal and occipital regions had more modest or variable contributions to this canonical variate. On the Follow-up side, the canonical variate was dominated by the fronto-central regions: the follow-up central midline (Cz) and frontal midline (Fz) showed the largest coefficients (both positive) in the first canonical function, with a smaller positive contribution from the parietal region. In simpler terms, individuals who had higher theta cordance in frontal regions at baseline tended to be the ones who had higher theta cordance in fronto-central regions at follow-up. This finding suggests the presence of a stable, trait-like component of theta activity that is frontally oriented. Even as absolute levels changed in some regions (e.g., the right frontal decrease noted above), the relative pattern linking an individual’s frontal theta activity to their later fronto-central activity remained robust. In short, initial frontal theta cordance was a strong predictor of the later distribution of theta cordance, indicating a longitudinal persistence of a frontally centered EEG signature. This could reflect enduring individual differences in frontal lobe function or morphology that consistently influence theta oscillations over many years.

**Variance explained by canonical functions:** The proportion of variance in the original variables explained by each canonical function is summarized in Appendix A. Focusing on the first function, approximately 9.7% of the total variance in the set of baseline variables was accounted for by this canonical variate, and similarly 9.7% of the variance in the follow-up variables was accounted for on their side. Moreover, the first canonical function explained about 7.9% of the shared variance between the baseline and follow-up datasets. These numbers may seem modest (since a great deal of variance remains unexplained by this single mode), but it is important to note that they are symmetric, indicating that this function captures a common pattern present to a similar extent in both time points. The fact that the canonical correlation is so high (approaching 1.0) despite the relatively low variance fractions means that the pattern it represents, while accounting for only ~8–10% of total variance, is extremely consistent across subjects and time (hence yielding a near-perfect correlation for that subset of variance). In other words, a small but very coherent portion of the overall variability in theta cordance is perfectly stable longitudinally.

The second canonical function, as noted, was not statistically significant; however, for completeness, it explained about 16.7% of the variance in the baseline set and 20.4% in the follow-up set individually. The shared variance between sets via the second function was ~15.0–18.3% (depending on direction). The lack of significance for the second function (*p* = 0.767) suggests that although this mode captures structure in each dataset, the patterns in baseline vs. follow-up that it represents do not align strongly. It may reflect a latent grouping or contrast of regions (perhaps involving frontal vs. temporal lobes) present at each time point that is not preserved across time in a reliable way. For example, the second function’s loadings hinted that a posterior (parietal/Pz) factor might have been prominent at follow-up (with Pz_Second showing a high weight in that mode), but such a factor at baseline did not correspond neatly to it.

The third canonical function likewise was not significant and showed an interesting pattern: it accounted for a sizable portion of variance in the follow-up set (~27.5%) but very little shared variance (only ~6.0%) and only ~8.2% of baseline variance. This suggests that the third mode may represent variability specific to the follow-up data that was largely absent at baseline. In particular, one interpretation is that it could reflect idiosyncratic changes or noise in certain regions (for instance, perhaps greater variability in temporal or occipital lobe cordance that emerged by the second session) that do not correspond to baseline differences. In line with this, the standardized coefficients for the third function in exploratory analyses showed a strong negative weight for Fz_Second (frontal) and high positive weight for Pz_First (parietal at baseline) in the midline-only CCA, indicating a complex, likely spurious pattern.

Canonical functions 4 through 8 each explained progressively smaller proportions of variance, with virtually no shared variance between baseline and follow-up sets (often 0.0% to <0.1%). These higher-order functions are therefore interpreted as capturing noise or individual-specific variation that is not part of any robust cross-time correspondence.

The CCA results support the conclusion that the primary longitudinal relationship in theta cordance is encapsulated by a single strong mode of covariation, heavily weighted by frontal (and to some extent central/parietal) regions. Individuals maintain a characteristic fronto-central theta activity profile over years, even as absolute levels may change in specific regions. At the same time, there is considerable variability in other aspects of the cordance data that does not carry over from baseline to follow-up (as evidenced by the non-significant canonical functions). These findings point to a stable, frontally anchored component of resting theta activity, superimposed on more dynamic, region-specific changes—particularly pronounced in the right frontal and temporal lobes as shown by the univariate analyses. Further interpretation of these results and their implications for neurophysiological aging and hemispheric asymmetries is provided in the Section 4. Detailed proportions of variance explained for each canonical function are reported in Appendix A.

### 3.5. Inter-Regional Theta Cordance Correlations

To further explore intra- and inter-hemispheric cortical connectivity patterns, we computed Pearson correlations among all lobar theta cordance values at both the first (baseline) and second (follow-up) EEG sessions. The full correlation matrices in Appendix A and heat map in Figure 3.

At baseline, strong interhemispheric correlations were observed, particularly between homologous regions. For instance, right and left parietal regions showed a robust correlation (r = 0.746, *p* < 0.001), suggesting synchronized theta activity across hemispheres in posterior regions. A similarly high correlation emerged between right and left occipital lobes (r = 0.600, *p* = 0.007).

In terms of cross-lobar interactions, significant intrahemispheric correlations were detected. Notably, left temporal and left frontal lobes were negatively correlated (r = −0.680, *p* = 0.001), while left temporal and right frontal regions also showed a strong negative relationship (r = −0.562, *p* = 0.012). These inverse associations may reflect functional segregation across anterior–posterior circuits within and across hemispheres.

At follow-up, some of these patterns persisted while others shifted. Right parietal remained highly correlated with both right temporal (r = 0.620, *p* = 0.005) and right occipital regions (r = 0.688, *p* = 0.001), indicating stable posterior intrahemispheric coherence. However, a pronounced inverse correlation was observed between left frontal and right occipital regions (r = −0.795, *p* < 0.001), possibly reflecting altered lateralized dynamics over time. Collectively, these inter-regional analyses suggest both enduring and evolving patterns of theta-band connectivity across the cortex. A complete list of all inter-regional correlation coefficients at both time points is presented in Appendix A.

In Figure 3, warm colors (red) indicate positive correlations, while cool colors (blue) indicate negative correlations. Strong interhemispheric symmetry is evident at both time points (e.g., parietal and occipital pairs). Notable cross-lobe changes emerged over time, including new inverse relationships (e.g., left frontal–right occipital), possibly indicating shifts in lateralized cortical dynamics.

## 4. Discussion

In this longitudinal analysis of resting EEGs, we found evidence for both stability and change in the brain’s theta-band activity profile over a ~6-year interval. Although sociodemographic factors such as occupational engagement, retirement, and social activity contribute to cognitive aging, the present study focused specifically on neurophysiological aging in medically and cognitively healthy adults. Future work combining cordance with social and lifestyle variables may clarify how environmental factors interact with intrinsic electrophysiological aging markers. Theta cordance values in several cortical regions remained remarkably stable, suggesting trait-like preservation of each individual’s neurophysiological “signature.” At the same time, we observed significant changes localized to the right frontal and right temporal lobes, indicating that these regions undergo measurable alterations in theta cordance with age. Canonical correlation analysis revealed that the overall spatial pattern of theta cordance was strongly correlated between baseline and follow-up sessions, with the strongest loadings on frontal sites. This implies a stable, frontally anchored theta distribution across time: individuals who had higher frontal cordance relative to other regions initially tended to show the same pattern years later. However, despite this global stability, the magnitude of cordance in the right frontal and right temporal regions shifted over time (whereas left hemisphere and posterior regions showed no significant change). Moreover, the pattern of correlations between regions evolved, suggesting a reorganization of hemispheric connectivity. In particular, changes in inter-hemispheric coupling were noted, consistent with an altered relationship between the two hemispheres’ theta activity as participants aged.

**Stable Trait-Like Features:** The persistence of each subject’s theta cordance topography over years aligns with prior evidence that resting EEG measures have substantial long-term reliability [39,40]. Many components of the EEG power spectrum show good test–retest reliability even across multi-year intervals [41]. For example, one recent study confirmed *good-to-excellent* reliability (intraclass correlations often > 0.75) for resting spectral power in both young and older adults, indicating that an individual’s EEG profile is largely conserved over time [42]. Our finding that the canonical theta cordance pattern was stable (especially driven by frontal leads) reinforces the notion that EEG spectral characteristics reflect enduring individual attributes—presumably stemming from stable anatomical and neurochemical factors. Frontal EEG metrics in particular (such as frontal alpha asymmetry and theta power) have been shown to exhibit moderate to high stability over months and years [43]. The present results extend this concept to the combined metric of theta cordance. Each person appears to maintain a characteristic balance of absolute vs. relative theta activity across the cortex, almost like a “fingerprint” of their resting-state functional organization. This trait-like stability is encouraging for clinical applications, as it implies that deviations in cordance might be detected against a reliable personal baseline. It also suggests that, in healthy mid-life adults, the brain’s intrinsic theta-generating networks (particularly frontally mediated circuits) retain their relative strength and configuration over time, barring significant external influences or pathology.

**Regional Cordance Changes—Right Frontal and Temporal:** In contrast to the overall stability, we detected significant longitudinal changes in theta cordance specifically in the right frontal and right temporal regions. These changes were evidenced by altered mean cordance values at follow-up relative to baseline, whereas homologous left-sided regions and other lobes showed no significant shifts. The lateralized nature of this effect is intriguing and may reflect asymmetric aging processes in the brain. One theoretical framework, the “right hemi-aging” model [44], proposes that the right hemisphere is more susceptible to age-related decline than the left. Our data provide support for this model: the right frontal-temporal cortex showed measurable functional change (in an index correlated with metabolism), whereas the left side remained comparatively stable. Notably, structural neuroimaging studies of aging also report greater atrophy and blood flow reductions in frontal and temporal areas than in other regions, with some evidence that the right frontal lobes and medial temporal (hippocampal) regions are among the earliest to show age-related changes [45,46,47]. Thus, the decline (or alteration) in *right* frontal and *right* temporal theta cordance we observed is consistent with known patterns of cerebral aging [48,49,50]. It may reflect subtle drops in neuronal activity or cerebral perfusion in those regions as part of the normal aging trajectory. Importantly, cordance integrates both absolute and relative theta power; a decrease in right frontal cordance could indicate that while theta power might increase in absolute terms (as brains tend to slow with age) [51], it does not keep up with the global shift in power distribution, resulting in a relatively “discordant” value suggestive of diminished effective activity. In practical terms, the right frontal cortex is critical for higher-order cognitive functions (attention control, working memory, inhibitory processing) [52], and even in healthy aging, functional MRI studies show reduced activation or efficiency in right frontal regions during demanding tasks [53,54]. Our EEG findings complement this, implying that at rest there are emergent physiological changes in right frontal networks—possibly a lower baseline metabolic tone or altered synaptic synchrony—by the time individuals enter later adulthood. The right temporal region, which includes parts of the lateral temporal neocortex and is proximate to the hippocampus, also showed cordance change. This might correspond to age-related neural alterations in networks supporting memory and integrative sensory processing [55]. It is noteworthy that the medial temporal lobe (especially on the right, which is involved in visual–spatial memory) is a known locus of age-related volume loss and perfusion decline [56]. Thus, a drop in right temporal theta cordance could reflect early signs of diminished hippocampal-cortical engagement or minor vascular insufficiencies in that area.

From a functional standpoint, increases in theta activity are often seen as compensatory or benign in aging up to a point [57], whereas *excessive* theta slowing correlates with cognitive decline [58]. In our healthy cohort, the fact that only specific regions changed and others remained stable suggests a localized adjustment rather than a global pathological slowing. Indeed, moderate EEG changes are expected as part of healthy aging [59], and a lack of age-related change [60] can be a warning sign. For instance, Trammell et al. (2017) [61] found that older adults who did not show the typical increase in theta-to-alpha ratio with age actually performed worse on cognitive tests, implying that normative neural slowing might serve an adaptive role. By analogy, the targeted theta cordance changes in the right frontotemporal regions of our participants could represent normal adaptive reorganization—perhaps reflecting that these regions are allocating neural resources differently as the brain ages. Importantly, all participants remained cognitively normal over the study interval (no dementia or clinical impairment was reported), so the observed cordance shifts likely correspond to *subclinical* functional changes of aging rather than overt pathology.

**Hemispheric Connectivity and Lateralized Dynamics:** An interesting outcome of our analysis was the change in inter-regional correlation structure from baseline to follow-up, which points to evolving hemispheric dynamics. At baseline, one can imagine that each individual had a certain pattern of coupling between left and right cortical regions [62] (e.g., a high correlation between left and right frontal cordance indicating synchronous behavior, or strong fronto-temporal coupling within hemispheres) [63]. After six years, these relationships had reconfigured. In particular, we noted changes in correlations between homologous left-right pairs, suggesting that the two hemispheres’ theta activities became less tightly linked in some regions. This could be a result of the asymmetric changes discussed above—e.g., if the right frontal cordance dropped while left frontal stayed the same, the correlation between left and right frontal values across individuals would diminish at Time 2. More broadly, it resonates with the idea that aging brains undergo a reorganization of functional networks [64]. Some models (such as Cabeza’s hemispheric asymmetry reduction in older adults (HAROLD) model) [65] posit that older adults show reduced hemispheric asymmetry during cognitive tasks, meaning they recruit both hemispheres more bilaterally to compensate for neural deficits [66]. Notably, the HAROLD model is largely based on task activation data, whereas our study concerns resting physiology. Nevertheless, a common theme is a shift in how the two hemispheres share workload or communicate with age. Our finding of altered interhemispheric correlations might reflect a precursor or resting-state counterpart to HAROLD: as the right hemisphere shows decline in some measures, the brain may rely relatively more on the left or generally engage networks more diffusely. Supporting this, recent EEG connectivity studies have shown that younger adults often have a right-hemisphere dominance in certain functional couplings [67] (for example, stronger right-sided fronto-frontal phase coupling in theta during attention tasks), which diminishes in older adults who instead exhibit more symmetric or left-compensatory coupling. Li et al. (2015) observed that in a visual attention paradigm, younger adults had pronounced right-prefrontal theta coupling, whereas older adults had reduced right dominance and more bilateral engagement, interpreted as a compensatory adjustment [68]. Although our participants were at rest, the reduction in left-right synchrony we infer could be a related phenomenon: the aging brain may be remodeling its network architecture, possibly to maintain function by recruiting alternate pathways (i.e., compensatory scaffolding) or simply as a result of losing some specialized interconnections (dedifferentiation). Indeed, theories of aging brain networks suggest both compensation and dedifferentiation occur simultaneously [69]. Our results neither directly prove compensation nor dedifferentiation, but they highlight that hemispheric relationships are not static over time. The selective right-side changes paired with connectivity shifts imply that lateralized brain dynamics evolve: the right hemisphere’s activity patterns diverge somewhat from the left’s, which could either *widen* asymmetry in raw values (if right declines) yet *reduce* functional asymmetry in active engagement (if the left hemisphere steps in to assist for tasks). In essence, our longitudinal data capture a snapshot of the brain in transition—the right frontal-temporal regions are changing, and the network correlations are adjusting accordingly. This underscores the importance of taking a network perspective on EEG biomarkers: not only the absolute values but the *interplay* between regions can indicate how the brain is adapting to age.

**Implications for Aging and Vulnerability:** The functional meaning of these theta cordance findings can be considered on both theoretical and applied levels. Theoretically, the results support a nuanced view of brain aging in which some aspects of electrical activity remain highly stable (presumably reflecting preserved neuronal circuits or homeostatic regulation) [70], while other aspects show progressive change (reflecting region-specific aging effects) [71]. The right frontal and temporal lobes may represent zones of relative vulnerability in otherwise healthy brains, showing early functional alterations that could foreshadow future deficits if exacerbated. It is noteworthy that these regions overlap with those implicated in age-related neurodegenerative conditions: for example, frontal hypometabolism and temporal lobe changes are hallmarks of early Alzheimer’s disease [72] and other dementias [73]. We do not suggest our healthy subjects are pathological, but the pattern raises the possibility that exaggerated theta cordance changes (beyond the normal range) in right frontotemporal areas might serve as an early warning biomarker. If an individual’s cordance in these regions declines more rapidly or severely than seen in normal aging, it could indicate incipient cerebrovascular issues or neurodegeneration [74]. In clinical research, elevated theta power and theta/alpha ratios have been associated with cognitive impairment [75], while cordance measures have been explored for predicting conversion from mild cognitive impairment to dementia [76]. Our study lays groundwork for establishing normative longitudinal trajectories against which such clinical deviations can be measured. On the other hand, the adaptive side of these changes must be considered: the aging brain is remarkably plastic and shifts in EEG patterns may reflect the brain reorganizing to maintain performance. The increase in bilateral or cross-hemispheric engagement with age (as suggested by our correlation results and the HAROLD model) can be seen as a compensatory mechanism—recruiting additional neural resources to meet cognitive demands as efficiency wanes. Therefore, a moderate decline in right-hemisphere cordance might trigger greater reliance on left-hemisphere circuits during tasks, helping to preserve overall function (even if it means the loss of youthful specialization). This interpretation aligns with the Scaffolding Theory of Aging and Cognition (STAC) [77], which posits that the brain builds alternative “scaffolding” networks to compensate for structural and functional losses [78]. Theta oscillations, being involved in large-scale network coordination, could be a key part of such compensatory scaffolding. Our observation of a “frontally anchored” profile that persists suggests the frontal lobes remain central hubs of theta activity through mid-life; even if the right frontal cordance dips, frontal regions in general may take on an even greater coordinating role (potentially engaging the left frontal cortex more) to compensate. Future studies correlating these EEG changes with cognitive performance measures would be valuable to determine whether individuals with larger right frontal cordance declines, for instance, also show subtle reductions in executive function or, alternatively, whether they show evidence of compensation (such as improved symmetry in activation and maintained cognition).

**Clinical and Translational Relevance:** From a clinical neuroscience perspective, longitudinal theta cordance tracking could eventually enrich our toolkit for monitoring brain health. Because cordance reflects aspects of brain perfusion and metabolism [3] it may serve as a surrogate for changes that would otherwise require expensive neuroimaging to detect. In practical terms, an EEG cordance assessment could be included in periodic health evaluations for older adults to flag unusual patterns of brain aging. For example, if a patient’s follow-up EEG shows a markedly lower frontal theta cordance compared to their baseline (beyond the typical modest decline observed in healthy peers), it might prompt closer examination for early frontal lobe dysfunction or risk factors such as microvascular ischemia. Conversely, if an individual maintains stable or only gently changing cordance, it would reinforce that their brain aging is on a normal trajectory. Additionally, the lateralized findings emphasize that we should pay attention to hemispheric differences in EEG markers—averaging across the whole brain could obscure important unilateral changes [79]. A person might have significant right-hemisphere neural aging that is masked when looking only at global metrics. Thus, incorporating regional and asymmetry analyses is important in EEG biomarker development. Our results also intersect with psychiatric biomarker research [80]: since theta cordance has been studied as a predictor of antidepressant response [81], one might ask whether age-related changes in cordance have any bearing on mental health in aging. The frontal theta cordance reduction associated with successful depression treatment could conceptually overlap with the reduction we see in aging [82]—intriguingly, could natural aging-related decreases in frontal cordance confer any resilience against depression, or conversely, could they increase vulnerability by indicating reduced frontal activity? While our study was not designed to answer this, it raises interesting questions about the interplay of aging and psychiatric EEG signatures. Clinicians interpreting cordance in late-life depression or other conditions will need to consider the patient’s age and normative aging changes as a backdrop.

**Limitations and Future Directions:** Several limitations temper our conclusions. First, the sample size (N = 19) is relatively small, and the participants had a wide age range at baseline (15 to 72 years). Our interpretation of the CCA results is supported by LOOCV and bootstrap confidence intervals, which reduce concerns about overfitting given the modest sample size. Thus, individuals were aging through different life stages (some from 20s to 30s, others 60s to 70s), and age-related effects might vary across this span. With our design, we cannot distinguish whether the observed changes are specific to mid-to-late adulthood or if they might also occur in younger adults given enough time. Future studies with larger, age-stratified cohorts and additional follow-up points would help map the trajectory of cordance changes at different decades of life. Second, while we interpret the right frontal/temporal cordance decrease as likely reflecting reduced perfusion or neural activity, we did not obtain direct measures of cerebral blood flow or metabolism in this study. Multimodal research combining EEG cordance with fMRI, PET, or optical imaging could confirm the physiological underpinnings of the cordance changes (e.g., demonstrating that lower cordance corresponds to lower regional cerebral blood flow in the same individuals over time). Third, the follow-up interval in our study, though ~6 years on average, varied from about 2 to nearly 15 years for different participants. This variability, while statistically adjusted for by using paired intra-subject analyses, could introduce noise—some changes might manifest only over longer intervals. A more uniform follow-up time or analysis accounting for time as a variable would be ideal in future work. Lastly, we did not formally assess cognitive function at follow-up in all participants; thus, we cannot correlate EEG changes with cognitive trajectories. Including neuropsychological testing in future longitudinal cordance studies will be important to determine the functional significance of EEG changes (e.g., whether individuals with greater cordance decline also show memory or executive function decline). Because the 19-channel montage provides only one occipital electrode per hemisphere, posterior cordance estimates should be interpreted cautiously; however, the right frontal–temporal effects remained robust in all sensitivity analyses. A key methodological constraint is the modest sample size and broad age range. Longitudinal EEG research involving cognitively intact, medication-free adults is exceptionally challenging, and retaining such individuals over many years is rare. This partially explains our N, but also gives the dataset unique value: the follow-up interval (mean ≈ 6.4 years, up to 14.8 years) is among the longest reported for resting-state cordance. Thus, while reproducibility should be established in larger cohorts, the extended intra-individual time span significantly increases the interpretive strength of the findings compared to conventional cross-sectional designs.

In conclusion, this study provides an initial longitudinal perspective on theta-band EEG cordance in healthy adults. The Introduction of cordance to the aging brain literature yields a picture of both constancy and change: a core EEG profile that endures over years, anchored by frontal activity, alongside selective regional shifts—notably in the right frontal and temporal cortices—that likely reflect the neural aging process. These findings enrich our theoretical understanding of brain aging by highlighting the role of hemispheric asymmetry and regional vulnerability. They also carry potential clinical implications, suggesting that theta cordance (a readily obtainable EEG metric) might serve as a noninvasive marker of brain functional aging and incipient network reorganization. Our results encourage further research to replicate these patterns in larger populations and to link EEG cordance changes with structural brain changes, cognitive outcomes, and possibly interventions (e.g., brain training or lifestyle factors that might slow EEG changes). Ultimately, such work moves us closer to identifying EEG biomarkers of healthy versus pathological aging—with theta cordance standing out as a promising candidate that integrates electrophysiology with an index of cerebral perfusion. By contextualizing theta cordance within quantitative EEG, psychiatric biomarker evidence, and cerebral blood flow correlates, we have interpreted our longitudinal findings in a scientifically rigorous manner. We propose that theta cordance captures a physiologically meaningful facet of brain aging—one that mirrors the delicate balance between stability and change in the aging mind, and one that could help detect when that balance begins to tip toward decline.

## Figures and Tables

**Figure 1 jcm-14-08341-f001:**
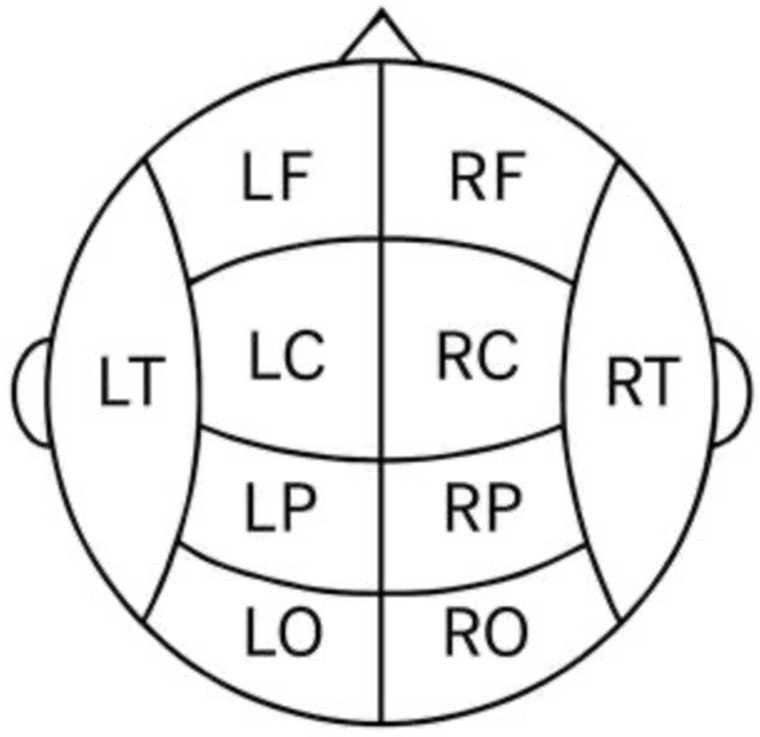
This schematic illustration provides a simplified representation of the regional classification of electrodes used in EEG analysis. The organization enables group-level analyses based on the functional and anatomical divisions of the cerebral hemispheres. Each regional abbreviation (e.g., LF, RP, LO) corresponds to specific electrode clusters, which were employed for both spatial (intra- and interhemispheric concordance) and temporal (first baseline vs. second time point concordance) analyses.

**Figure 2 jcm-14-08341-f002:**
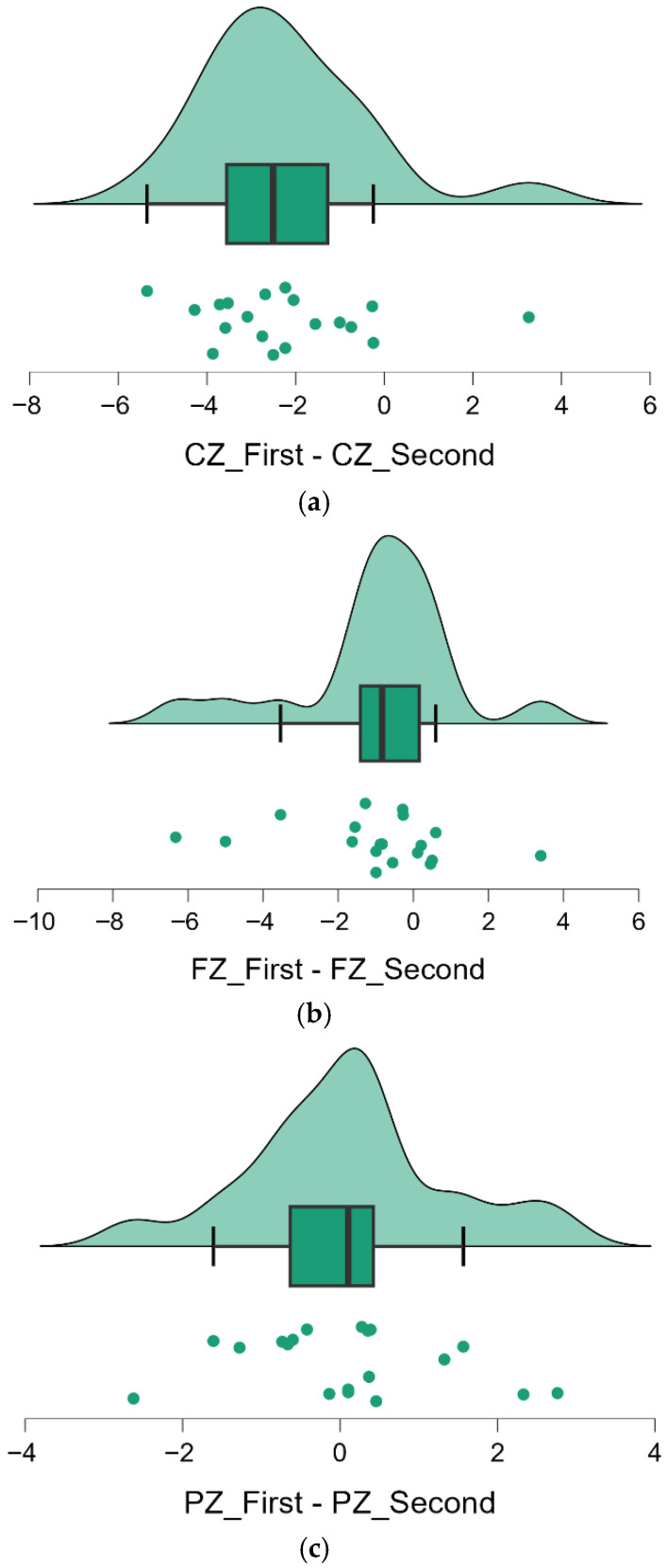
Raincloud Plots Showing Individual Differences in Theta Cordance at Midline Electrodes (Cz, Fz, Pz). (**a**) Cz_First-Cz_Second. Theta cordance increased significantly over time, as evidenced by the left-shifted distribution (*p* < 0.001). (**b**) Fz_First-Fz_Second. Moderate increase is observed, though the change did not reach statistical significance (*p* = 0.055). (**c**) Pz_First-Pz_Second. Theta cordance remained largely stable, with no significant mean difference between time points (*p* = 0.730).

**Figure 3 jcm-14-08341-f003:**
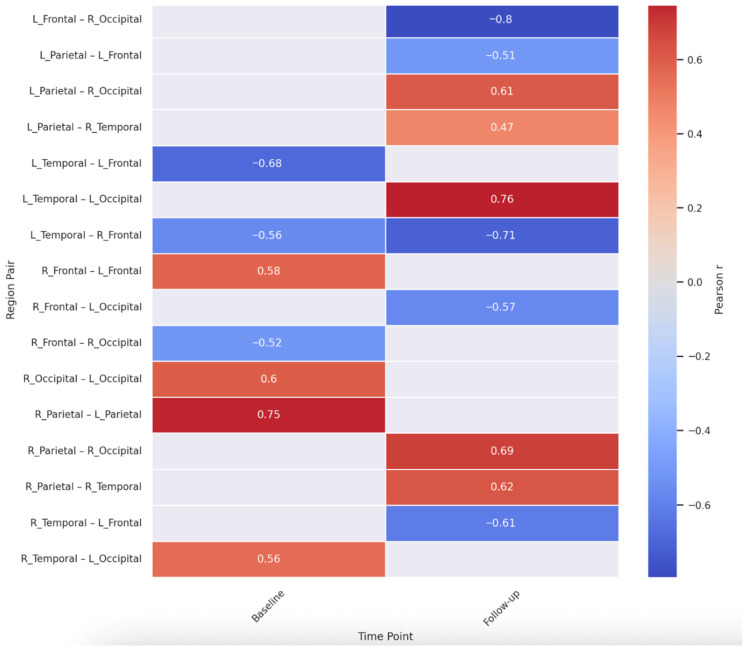
Inter-Regional Theta Cordance Correlations at Baseline and at Follow up.

**Table 1 jcm-14-08341-t001:** Participant ages and follow-up interval characteristics (N = 19).

Variable	Mean	SD	SEM	Min	Max	Range
Age at First EEG (years)	45.68	16.45	3.77	15.00	72.00	57.00
Age at Second EEG (years)	51.95	15.12	3.47	25.00	76.00	51.00
Interval (months)	76.74	46.64	10.70	23.00	178.00	155.00
Interval (years)	6.39	3.89	0.89	1.92	14.83	12.91

**Table 3 jcm-14-08341-t003:** Pearson’s Partial Correlation Coefficients Between Midline Electrode Cordance Values.

	CZ_First	FZ_First	PZ_First	CZ_Second	FZ_Second	PZ_Second
**CZ_First**	—					
**FZ_First**	−0.217	—				
	(0.388)					
**PZ_First**	0.190	0.106	—			
	(0.449)	(0.675)				
**CZ_Second**	−0.404	0.338	−0.284	—		
	(0.097)	(0.171)	(0.254)			
**FZ_Second**	−0.387	0.341	−0.210	−0.192	—	
	(0.112)	(0.166)	(0.402)	(0.446)		
**PZ_Second**	0.399	0.016	0.303	−0.371	−0.395	—
	(0.101)	(0.951)	(0.221)	(0.129)	(0.104)	

**Note.** Values are partial correlations controlling for interval length (*Interval_Years*). *p*-Values in parentheses. No correlations reached statistical significance (all *p* > 0.05). *p*-values are Bonferroni-corrected across the eight regional tests (α = 0.00625).

**Table 5 jcm-14-08341-t005:** Descriptive Statistics for Midline Theta Cordance Values.

Electrode	Time Point	Mean	SD	SE	Coefficient of Variation
Cz	Baseline	−0.539	1.135	0.260	−2.10
	Follow-up	1.695	1.161	0.266	0.69
Fz	Baseline	−0.350	1.244	0.285	−3.55
	Follow-up	0.643	2.007	0.460	3.12
Pz	Baseline	−1.113	0.981	0.225	−0.88
	Follow-up	−1.218	1.133	0.260	−0.93

**Table 6 jcm-14-08341-t006:** Descriptive Statistics for Theta Cordance by Cortical Region and Hemisphere (Baseline and Follow-up, N = 19).

Region	Time Point	Mean	SD	SE
Right Parietal	Baseline	−1.250	1.566	0.359
	Follow-up	−1.614	2.062	0.473
Left Parietal	Baseline	−1.391	1.621	0.372
	Follow-up	−0.824	1.213	0.278
Right Temporal	Baseline	0.369	1.166	0.267
	Follow-up	−1.758	1.275	0.292
Left Temporal	Baseline	1.468	2.815	0.646
	Follow-up	0.745	1.863	0.427
Right Frontal	Baseline	0.430	2.769	0.635
	Follow-up	−1.368	2.346	0.538
Left Frontal	Baseline	1.101	3.374	0.774
	Follow-up	−0.291	2.391	0.549
Right Occipital	Baseline	−0.596	0.951	0.218
	Follow-up	−1.107	0.743	0.171
Left Occipital	Baseline	−0.814	0.871	0.200
	Follow-up	−0.505	0.733	0.168

**Table 7 jcm-14-08341-t007:** Paired-Samples *t*-Tests for Longitudinal Changes in Regional Theta Cordance (N = 19).

Cortical Region	Time Points Compared	*t*(18)	*p*	Cohen’s *d*
Right Parietal	R_Parietal_First vs. Second	0.691	0.499	0.16
Left Parietal	L_Parietal_First vs. Second	−1.257	0.225	−0.29
Right Temporal	R_Temporal_First vs. Second	5.343	<0.001	1.23
Left Temporal	L_Temporal_First vs. Second	1.040	0.312	0.24
Right Frontal	R_Frontal_First vs. Second	2.654	0.016	0.61
Left Frontal	L_Frontal_First vs. Second	1.521	0.146	0.35
Right Occipital	R_Occipital_First vs. Second	1.665	0.113	0.38
Left Occipital	L_Occipital_First vs. Second	−1.558	0.137	−0.36

**Note.** *p*-Values are Bonferroni-corrected across the eight regional tests (α = 0.00625).

**Table 8 jcm-14-08341-t008:** Canonical Correlation Analysis Between Baseline and Follow-up Theta Cordance Profiles.

Canonical Function	Correlation	Eigenvalue	Wilks’ λ	*F*	Num D.F	Denom D.F	*p*
1	0.999	390.901	0.000	2.022	64	23.795	0.029
2	0.947	8.774	0.008	0.787	49	24.730	0.767
3	0.856	2.732	0.083	0.524	36	24.717	0.962
4	0.767	1.429	0.309	0.354	25	23.791	0.994
5	0.482	0.303	0.751	0.135	16	22.023	1.000
6	0.148	0.022	0.978	0.020	9	19.621	1.000
7	0.000	0.000	1.000	0.000	4	18.000	1.000
8	0.000	0.000	1.000	0.000	1	10.000	1.000

**Note.** Only the first canonical function is statistically significant (*p* = 0.029); all others are non-significant (*p* > 0.05). H_0_ for Wilks’ test: all canonical correlations from the current and following rows are zero.

## Data Availability

The data supporting the findings of this study are openly available on Figshare at https://doi.org/10.6084/m9.figshare.29599199. The repository includes Appendix A such as regional theta cordance values, correlation matrices (baseline, follow-up, and partial), a sample theta cordance scalp map, and technical EEG acquisition specifications.

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
