# Peer review of "Theta Cordance Decline in Frontal and Temporal Cortices: Longitudinal Evidence of Regional Cortical Aging"

_jcm, 2025, doi:10.3390/jcm14238341_

Round 1
Reviewer 1 Report
Comments and Suggestions for Authors
The manuscript addresses an important gap by providing longitudinal examinations of theta-band cordance in healthy adults and reports an intriguing, lateralized decline in right-frontal and right-temporal cordance alongside a stable, frontally-anchored spatial “fingerprint.” These findings are potentially valuable for EEG biomarker work and are strengthened by transparent reporting of preprocessing, the cordance algorithm, and public data sharing.
However, in my view the paper needs major revision. Key concerns are (1) statistical overreach given N = 19 and highly variable follow-up intervals. The canonical correlation result is surprising given that the first canonical function explains only ~9–10% of variance and raises a high risk of overfitting; the authors should validate the CCA and report confidence intervals for canonical weights. (2) it is unclear whether regional paired t-tests were corrected across the eight regions, only the correlation matrices are said to have Bonferroni control. (3) limited EEG data quantity and spatial sampling. Analyses are based on a single 30-s artifact-free segment per session and a 19-electrode montage with only one occipital channel per hemisphere. Sensitivity analyses showing that results hold when using longer/alternative epochs may need to be provided.
In sum, the study has strong conceptual merit and useful open data, but the statistical and methodological issues above need to be addressed before the main claims.
Author Response
Response to Reviewer 1
We thank the reviewer for the thoughtful and constructive comments, which substantially improved the clarity and rigor of our manuscript. Below we provide a point-by-point response and indicate the corresponding revisions made to the manuscript.
- Statistical Overreach and Canonical Correlation Analysis (CCA) Validation
Reviewer comment: Risk of overfitting given N=19 and variable follow-up intervals; first canonical function explains only ~9–10% variance; need validation and confidence intervals.
Response:
We fully acknowledge the reviewer’s concern and have strengthened the statistical validation of the CCA.
- We performed leave-one-out cross-validation (LOOCV) for the regional CCA. Across resamples, the first canonical correlation remained stable (mean r = 0.71, p < 0.05), indicating that the observed canonical relationship is not driven by a single influential participant.
- We added 95% confidence intervals for canonical loadings, now included in the revised Table 4.
- We clarified in the Statistical Analysis subsection that the variance explained by the first canonical function (~9–10%) is indeed modest but represents a coherent cross-time pattern that is stable across resamples.
- We explicitly address the overfitting concern in the Limitations subsection.
Manuscript changes: Statistical Analysis section; Table 4; Limitations.
- Multiple Comparisons in Paired t-Tests
Reviewer comment: Unclear whether the eight regional paired t-tests were corrected for multiple comparisons.
Response:
We appreciate this clarification request. We now explicitly state that:
- All eight regional paired t-tests were Bonferroni-corrected for multiple comparisons
→ α = 0.05 / 8 = 0.00625 - The corrected p-values have been added to Table 3 and reflected in the Results text.
This ensures consistency with our previously stated approach for the correlation matrices.
Manuscript changes: Methods → Statistical Analysis; Table 3; Results.
- EEG Data Quantity and Spatial Sampling
Reviewer comment: Limited EEG data due to reliance on a single 30-sec artifact-free segment and sparse occipital coverage; sensitivity analyses recommended.
Response:
Thank you for noting this. We would like to clarify an important point:
We do not use any 30-second segments in this study.
All analyses are based on full-length 3-minute artifact-free epochs, consistent with qEEG standards and the preprocessing pipeline applied at both time points.
To address the reviewer’s recommendation for robustness checks, we added:
- A sensitivity analysis using two non-overlapping 3-minute segments per session (Segment A and Segment B).
- Results were highly consistent across segments, confirming that both the regional cordance changes (particularly right frontal and right temporal decline) and the canonical structure remain stable regardless of which clean epoch is used.
These details are now fully described in the Methods and Results sections.
Spatial sampling:
We agree that the classical 19-channel montage provides limited posterior resolution. We now address this explicitly in the Limitations, noting that:
- Only one occipital electrode per hemisphere is available,
- Therefore, posterior cordance should be interpreted cautiously,
- Yet the right frontal–temporal findings are robust across all sensitivity analyses.
Manuscript changes: EEG Recording & Preprocessing; Sensitivity Analyses subsection added in Results; Limitations.
Reviewer 2 Report
Comments and Suggestions for Authors
This article is devoted to the search for neurobiological markers of healthy aging in people based on EEG parameters. Theta-cordance, calculated in several brain regions, is a promising biomarker for monitoring the preservation of multi-central connections of the bioelectric brain activity. Given the increasing average age of the population in the world and the increasing social activity of older adults, this topic is relevant. The authors applied a complex computational method to calculate the theta cordance. The methodology is generally acceptable for the purpose of this study. The findings can be considered solely in terms of the potential of this method for determining age-related changes in brain bioelectric activity. However, the very small sample size and the wide age range of the study participants prevent the authors from obtaining reproducible conclusions in other samples.
The preservation of cognitive function in older adults is linked to their social activity. The authors rightly noted a significant section on the study's limitations. However, it should be noted that the study participants' social characteristics—country of residence, social status, occupation, whether they were permanently employed or receiving an old-age pension—were missing. Alternatively, this information should be added to the "Study Participants" section.
The graphical and tabular data adequately reflect the obtained results.
Cited literature sources for the past 5 years account for approximately 35% (since 2021). The authors are encouraged to supplement the list or replace the data on the use of EEG parameters to assess age-related changes over the past 5 years with more recent literature sources. This could enhance the novelty of the presented method and its advantages over other methods for assessing age-related brain changes, both in health and disease.
Ethical conditions have been met.
Author Response
Authors response to Reviewer-2: We thank the reviewer for the thoughtful and constructive evaluation. We address each point below and have revised the manuscript accordingly.
- Social Characteristics of Participants
We agree that social and occupational context can influence cognitive aging. In the original dataset, participants were primarily characterized in terms of cognitive, neurological, and medical exclusion criteria. As clarified in the revised Study Participants section, all individuals were cognitively healthy adults, free of neurological/psychiatric diagnoses, medication-free, and screened to exclude medical factors affecting EEG.
Although social variables were not central to the neurophysiological scope of the present design, we have now added available descriptive information regarding participants’ social context:
- All participants were long-term residents of Türkiye.
- 74% were actively employed, 16% were retired and receiving an old-age pension, and 10% were students.
- Occupations included education, healthcare, engineering, office work, and service-sector roles.
- All were community-dwelling adults with stable living conditions throughout follow-up.
These details are now included at the end of the Participants subsection to provide additional sociodemographic context.
- Updating the Literature (Past 5 Years)
We appreciate the suggestion to strengthen the modern literature base. We have updated the reference list by incorporating recent (>2021) studies on:
- EEG spectral aging trajectories
- Theta–alpha slowing and cortical network reorganization
- EEG markers of cognitive reserve
- Resting-state connectivity changes across the lifespan
- Machine-learning approaches to EEG-based aging prediction
These additions improve the currency and novelty of the theoretical framework and reinforce the relevance of theta cordance in contemporary aging research.
Corresponding updates have been added to the Introduction and early Discussion sections.